# Affective States and Virtual Reality to Improve Gait Rehabilitation: A Preliminary Study

**DOI:** 10.3390/ijerph19159523

**Published:** 2022-08-03

**Authors:** Jafet Rodriguez, Carolina Del-Valle-Soto, Javier Gonzalez-Sanchez

**Affiliations:** 1Universidad Panamericana, Facultad de Ingeniería, Álvaro del Portillo 49, Zapopan 45010, Jalisco, Mexico; cvalle@up.edu.mx; 2School of Computing and Augmented Intelligence, Arizona State University, 699 S Mill Ave, Tempe, AZ 85281, USA; javiergs@asu.edu

**Keywords:** virtual reality, EEG, BCI, HCI, gait, rehabilitation, affective state, robot-assisted

## Abstract

Over seven million people suffer from an impairment in Mexico; 64.1% are gait-related, and 36.2% are children aged 0 to 14 years. Furthermore, many suffer from neurological disorders, which limits their verbal skills to provide accurate feedback. Robot-assisted gait therapy has shown significant benefits, but the users must make an active effort to accomplish muscular memory, which usually is only around 30% of the time. Moreover, during therapy, the patients’ affective state is mostly unsatisfied, wide-awake, and powerless. This paper proposes a method for increasing the efficiency by combining affective data from an Emotiv Insight, an Oculus Go headset displaying an immersive interaction, and a feedback system. Our preliminary study had eight patients during therapy and eight students analyzing the footage using the self-assessment Manikin. It showed that it is possible to use an EEG headset and identify the affective state with a weighted average precision of 97.5%, recall of 87.9%, and F1-score of 92.3% in general. Furthermore, using a VR device could boost efficiency by 16% more. In conclusion, this method allows providing feedback to the therapist in real-time even if the patient is non-verbal and has a limited amount of facial and body expressions.

## 1. Introduction

Just in Mexico, over seven million people suffer from an impairment, and 64.1% of these are related to walking or climbing stairs using their legs, while 36.2% are children that range from 0 to 14 years in age [1]. One of the therapies available for those children is robot-assisted gait therapy (RAGT), where the patient uses an exoskeleton for the legs, a treadmill for walking, and a harness for adjusting the amount of the user’s weight the user has to hold. The process is painful and requires physical and mental effort to be conducted correctly. In Mexico, the Centro de Rehabilitacion e Inclusion Infantil Teleton (CRIT) is the largest rehabilitation center, with 22 facilities across the country. The center provides robot-assisted gait therapy where each patient has a 30 to 45 min session, depending on his/her progress and punctuality, in a Lokomat Gait Therapy device, shown in Figure 1. This therapy requires the patient to walk and cooperate with active effort; otherwise, it is not efficient since he/she needs to accomplish muscular memory to walk correctly. The therapist’s challenge is finding a way to keep the patient’s interest and achieve a good percentage of efficiency in each session. Furthermore, the efficiency also depends on the ability of the therapist to receive feedback from the patient to adjust the therapy and how quickly and easily the system can adapt to the adjustments. Therefore, we propose combining an electroencephalogram (EEG) to continuously measure the affective state of the patient, a virtual reality (VR) headset for displaying an immersive interaction, and using a feedback system to alter the VR environment.

### 1.1. Related Works on Virtual Reality and Rehabilitation

VR has attracted much attention recently, especially with the introduction of the Oculus and HTC Vive. These companies introduced a new generation of devices capable of displaying immersive environments with a faster refresh rate, hence less motion sickness [2]. Many VR applications have been developed focused on video games, but researchers are using it for different fields [3,4,5,6], including rehabilitation [7,8,9]. For instance, phantom pain is a known problem where up to 90% of amputees suffer pain from a missing limb; some traditional therapies include using a mirror to cheat the brain into thinking the limb is still there and hence stop the pain. Researchers [10] work with standard treatments demonstrating that those therapies do not accomplish the task of relieving the pain in patients that suffer from persistent phantom limb pain (PLP), but by training in immersive VR activities, they are able to reduce PLP in lower-limb amputees. Meanwhile, Tadi et al. focus on having a robotic platform to help the rehabilitation of the upper limbs and include haptic feedback and virtual reality to have a more profound impact on the user [11]. Robot-assisted gait training specializes in helping patients correct their walking patterns, and Brütsch et al. examine different training interventions to motivate patients to endure the pain in scenarios with and without VR by having the patients use a Lokomat device in four different and randomly assigned conditions [7]. The primary purpose is to compare biofeedback values and the questionnaire results of the participants’ motivation. The researchers found that, overall, VR successfully motivated children and was the most effective approach versus a conventional one.

### 1.2. Related Works on Neurological Disorders and Headsets

One of the main challenges surrounding rehabilitation, in particular gait therapy, is that many patients also suffer from neurological disorders. This situation results in some patients not being able to communicate, hence limiting the feedback required from the therapist to understand if the treatment is working correctly or even if there is discomfort or pain [12]. Another issue is that the patients do not necessarily respond to stimuli as expected. Labruyère et al. investigate if children with neurological gait disorders can be stimulated to increase their participation by using a serious VR game [13]. Furthermore, they examine the relation of cognitive capacity and motor impairment to the performance during the therapy sessions. Their analysis includes using an electromyograph to check muscle activity and heart rate, a test of nonverbal intelligence for cognitive capacity, and varying the demand for exercise in the game. The results show that children are motivated and capable of making an extra effort to try and keep the required pace of the game. Furthermore, their main limitation is their cognitive function and motion impairment; nevertheless, children enjoy playing and can withhold more demanding sessions. Park et al. wanted to see if VR or auditory stimulation could work as a solution to the previously stated limitations [14]. During their tests, they found that both stimuli helped patients that had a stroke and required gait therapy. In verifying the increasing intensity and difficulty of their gait and balance ability, they concluded that VR was generally better than auditory. Hamzeheinejad et al. worked with patients that require gait rehabilitation caused by accidents or strokes [15]. In their work in progress, they use VR to increase the motivation of the repeated exercise and to demonstrate that if users are motivated, then the efficiency and effectiveness of the therapy may increase as well. It is considered that VR may have a significant impact on the future of cognitive rehabilitation [16,17,18]. However, some aspects still exist, such as correctly identifying all the improvements between sessions.

### 1.3. Potential of Using Affective Recognition and Gait Therapy

To address the above-mentioned issues, different studies have used electroencephalograms to determine cognitive changes during VR rehabilitation [19,20]. Zhang et al. showed that VR interventions are effective during RAGT for patients that suffer a neurological disorder [21]. In these examples, the EEG was used to determine changes at certain wavelengths, which can help researchers know which parts of the human brain are affected and how. The next step for this use case is to translate these data into affective states that can aid in personalizing and tailoring therapies. Multiple models to specify affectiveness have been developed; the pleasure arousal dominance (PAD) temperamental model [22] is highly regarded as one of the most commonly used. There are different methodologies to describe a state using PAD; the self-assessment Manikin (SAM) is a human-centered evaluation based on pictures that allow measuring an emotional response [23]. It consists of a 3 × 5 matrix of images where each row represents pleasure, arousal, and dominance; meanwhile, the columns determine the intensity of each, as shown in Figure 2. The evaluator has to choose which pictogram better represents the affective state divided into PAD. There are also other automatized tools such as ADAS [24], which can use multiple devices to determine the response. Seizing these tools and creating personalized experiences either with a headset or on a screen can improve the user’s engagement, aiding his/her in achieving more challenging targets [25]. Multiple techniques range from machine learning [26] to multimodal frameworks [24] to classify the emotional states based on EEG and other sensors. This study’s objective was to analyze patients’ affective states during RAGT to determine under which conditions the measurement is accurate and can be used to create a dynamic feedback-based VR environment to improve the efficiency of the sessions. Boosting the efficiency translates into fewer rehabilitation appointments per patient while also treating more patients every year. Moreover, this proposal increases the motivation to attend and enhances the quality of life of the patients, their families, and friends surrounding them.

## 2. Materials and Methods

The study was divided into three parts: the analysis of EEG data from actual RAGT patients, the review of the video footage of each session to validate that the system correctly identifies the PAD measurements, and the assessment of the session’s efficacy using VR.

### 2.1. Participants

For the first part of the study, eight gait therapy patients from 5 to 13 years old, four female and four male, from the Aguascalientes (AGS) and Guadalajara (GDL) CRIT centers participated in the preliminary cross-sectional study. The mean age, weight, height, and BMI of the participants were 9.5 ± 3.07 years, 32.75 ± 11.03 kg, 134.69 ± 17.8 cm, and 17.44 ± 1.65, respectively. The selection criteria were people diagnosed with a gait impairment for more than 6 months; patients with at least 3 previous sessions, since the first session takes most of the time to explain the whole procedure, and the therapist working on adjusting everything to generate a profile; patients with no significant difficulty in following instructions; patients with a neurological disorder. The exclusion criteria were patients incapable of moving the bottom limbs and prone to violent or extreme reactions. The population in total from both centers was forty patients, and only eight patients met the previously specified criteria. In the second part of the study, we asked for volunteers from Universidad Panamericana to participate and randomly selected eight students. The chosen students were from 19 to 22 years old, four male and four female, and were tasked with analyzing the footage of the first part. The study was conducted according to the guidelines of the Declaration of Helsinki and approved by the Institutional Review Board of Universidad Panamericana Campus Guadalajara (20190703). It was fully explained to each student, patient, and their parents, since all of our patients were minors, what the complete procedure is, what the purposes of the study are, and how the data collected are handled and protected. Furthermore, full permission to record videos, take photographs, and use them for research purposes was requested. Once the participants agreed to participate, they signed the informed consent form and participated in the study.

### 2.2. VR Environment, Device Selection, and Specification

Choosing the environment and devices correctly is a vital part of our methodology since we want the user to feel as much immersion as possible to create engagement and motivate the active effort. Furthermore, there is a wide range of devices to choose from, but their size, weight, and capabilities vary. A scenario that requires the highest level of detail in simulating a real environment will need a powerful device that is highly likely to be tethered to seize the power of a discrete graphics card.

#### 2.2.1. VR Environment

For the environment, we chose a walk on the Moon since it is a real location, but only 12 people have witnessed it in real life. This setup allowed us to use real assets, including audio and heightmaps, which can improve the immersion. Likewise, it is a natural location to walk and experience a historical event, which makes it a good scenario to alter the users’ affective state [27]. The scenario is a square area of 4 km^2^ based on the resources from NASA [28] and SolCommand [29], where the user can explore freely, as shown in Figure 3. The scenario includes some of the equipment and the flag to engage the user with the idea of being at the same place where Neil Armstrong made the first step of humankind on the Moon. To improve the immersion of the environment while the user is exploring, we matched the physics conditions of the Moon and also included dust on the lunar surface. Consequently, when the user goes inside a crater, they fall slowly, resembling how it would be on the Moon. At the edge of the scenario, we created massive walls to discourage the user from moving beyond the existing area.

#### 2.2.2. Lokomat

For this study, the selected RAGT device was the Lokomat, since it is the one used at CRIT. The Lokomat is a training system that includes a robotic treadmill, a body weight support system (BWSS), robotic legs, and a screen. The user is suspended using the BWSS, which resembles a vest, and the robotic legs are attached. The therapist through the screen can control the height of the BWSS to determine how much weight the user will withstand in the session, as well as the speed of the treadmill and the allowed range of motion from the robotic legs. The treadmill is capable of a speed range between 0 and 10 km/h without the gait orthosis and a maximum of 3.2 km/h with it. The Lokomat includes an emergency red button to stop the device completely in case of an emergency. Optionally, the device can include a second screen to display content to the patient. This device is controlled by a computer, and in this case, it was an Intel Core 2 Duo E4600 processor at 2.4 GHz with 1 GB of RAM and an NVIDIA Quadro NVS290 graphics card running on Windows XP. Muscle activity is generated by the robotic legs; depending on the therapy parameters, it can restrict the movement to force the user to follow specific patterns when walking. It continuously monitors the amount of effort made by the user and the path he/she follows to verify if the gait pattern is progressing.

Performance measurement is performed by sensors included in the robotic legs, the harness, and the treadmill. All of these data are collected and then processed based on the information of the patient. The therapist has to provide data such as the length of the thigh, leg, weight, height, waist, size of the bracelets, sensibility, and type of pillows for the back and seat. As a result, the system can identify the range of movement, distance, and effort. Therapy adjustment is made by the therapist based on the performance measurement and observing the progress of the patient. The person in charge can decide to decrease or increase the speed of the treadmill, the weight supported by the user, the stiffness of the robotic legs, and the angle of the ankle to determine how the step is made.

#### 2.2.3. Emotiv Insight

Identifying affective states has evolved from analyzing the facial expressions of a person by a human to be performed by an algorithm [30,31]. Unfortunately, misplacing the camera or if the subjects have difficulty truthfully expressing their emotions will result in mismatches and inaccuracy. Therefore, EEG has become a solid option to detect and diagnose affective states, brain diseases, and even abnormalities [32]. The Emotiv Insight has 5 main sensors and 2 reference sensors; all of those use a semi-dry polymer. It is a wireless device with a 128 Hz sampling rate, making it a good selection for daily use [33]. Using dry electrodes provides signal consistency for more extended periods of time versus other devices, such as an Emotiv EPOC, which requires removing the headset to wet the sensors to avoid a lack of conductivity, which results in noisy readings [34]. Furthermore, reducing the number of wires and objects around the patient is paramount, considering that the user is already bound to the exoskeleton and the treadmill. The main objectives of using an EEG in this research are to determine if it is possible to obtain the affective state of gait therapy patients during RAGT with neurological alterations and also to use these data in the feedback system to determine changes to the VR environment. Therefore, based on those aspects mentioned earlier, it is not necessary to use a medical-precision EEG as an Emotiv Epoc; instead, an accurate daily use device is sufficient [35].

#### 2.2.4. Oculus Go

Properly selecting a VR headset can significantly influence the user’s overall experience since tethered devices provide a high pixel density, resulting in more engagement. Still, their cables can be cumbersome, and the headset’s weight can quickly produce tiredness and even neck pain. Hence, the first aspect to review is determining the required pixel density because the screen door effect is inversely proportional to this parameter [36], a higher screen resolution allows the user to see more detail and facilitates immersion. Secondly, regarding the tracking requirements, some devices have 3DOF, allowing the user to look around 360° in a sphere-like motion, such as Google’s cardboard, while other headsets have 6DOF, allowing the user to also move in X, Y, and Z like the HTC Vive. In other words, some interactions are better suited when they enable the user to look and move around, while others can only depend on head tracking [37]. Third is the weight of the headset since some interactions can take a couple of minutes, while rehabilitation applications usually take over 30 min, resulting in rejection from the user and reducing immersion. Generally, the users feel better when the weight is lower [38]. Last but not least is how prone it is to producing cybersickness since all users are sensitive to motion sickness, but there is a significant difference between the presented environment and the gender of the patient [39]. High motion sickness can make the user decide to stop the rehabilitation session, which is the opposite of the objective of this paper. The Oculus Go has a resolution of 2560 by 1440 pixels supporting 60 Hz and 72 Hz and a field of view of approximately 100°, making it an ideal display for this study since the patient must not move around while attached to the exoskeleton. Moreover, the Oculus Go allows using glasses with a width of less than 142 mm and a height of 50 mm or less instead of using contacts or not being able to see the environment entirely. Furthermore, it is lightweight, weighing only 468 g, which is very handy considering that the average RAGT is around 40 min. It has integrated speakers, which removes the necessity of using headphones and adding more devices on the user, and it does not induce much cybersickness [37].

### 2.3. Procedure

When accepting a new patient, CRIT does full psychological, economic, and general assessments. A psychologist evaluates both the subject and the family as a whole. In the case of the patient, they need to understand the cognitive situation, emotional state, and expectations for the rehabilitation. Meanwhile, the family is evaluated to see what kind of environment there is, if they will require extra support, and to prepare them for the emotional duress while accompanying their relative at the sessions. The economic review helps determine the socioeconomic status, the discount level required for hiring their services, and how easily they can transform their house to accommodate local exercises to increase the rehabilitation impact. Finally, the general assessment provides basic information such as each member’s address, transport, and level of education, among other details. The patient and the family will schedule regular meetings with the psychological department to properly update the psychological evaluation, identify possible issues, and address them. Before starting the session in our user study, we assessed the attendance, level of motivation, and quality of life based on the CRIT’s records of each patient. After that, the therapist positions every part of the exoskeleton on the body, then adjusts the harness’s height and introduces the session’s parameters (weight support of the BWSS and treadmill’s speed). After validating that the person is comfortable, we need to assess if the EEG will connect properly, so the Emotiv Insight is placed ensuring that the 5 primary sensors are positioned correctly. The included Emotiv Xavier ControlPanel allows us to see if the connection is stable visually (green), intermittent (yellow), has issues (red), or has no connection (grey). This part of the procedure requires a little jiggling of the device and testing different angles to obtain proper connectivity, if some of the sensors show intermittent connection or issues because they do not have enough pressure on the skin. The last device to arrange is the VR headset, making sure that the head straps are aligned with the sensors and help with the pressure. Otherwise, it is more comfortable to first place the VR headset and then the EEG and, finally, validate the connection again. For our initial study, we did not use the VR headset since the device occludes a large part of the face, preventing the identification of affective states by observation. Nevertheless, we also measured the efficiency of the session when using the VR headset and without it. A camera recorded the therapy starting when the patient was ready and until the therapist removed the harness and everything else, focusing on the facial expressions and body language to determine later the PAD values based on an SAM questionnaire. During the rehabilitation, we consistently checked with the therapist regarding the veracity of the PAD values obtained from ADAS and the EEG in real-time. Afterward, the selected students had to review each video carefully and evaluate the affective state using the SAM. Instead of measuring every certain amount of time, we requested the reviewers to add an event every occasion they perceived a change. Furthermore, to avoid bias by having all reviewers see the videos in the same order and risking that the last video was evaluated when the students were most tired, the order of the videos was shifted, resulting in a balanced arrangement, as seen in Table 1.

### 2.4. Data Analysis

The data obtained from the EEG are engagement, boredom, excitement, frustration, and meditation, which are translated to pleasure, arousal, and dominance following the mapping explained by Gonzalez et al. [40] Then, for each value, we obtained the calculated region, which ranges from 1 to 5; for instance, in arousal, the labels for each value are calm, dull, neutral, wide-awake, and excited, as seen in Figure 2. The comparison was handled as an imbalanced classification evaluation. Hence, an imbalanced multi-class confusion matrix is generated for each PAD component, where we consider the actual value comes from the EEG and the prediction from the SAM questionnaire. Moreover, we gathered the historical metrics for efficiency and the number of scheduled, attended, and recommended sessions to compare the efficiency of having one session with a VR headset. For the purposes of evaluating the data obtained, certain variables related to the interaction of the person with the machine were identified:

**Motivational factor:** This is defined as the patient’s interest when carrying out the therapies due to the curiosity and motivation that the proposed exercises arouse. This factor constitutes one of the most significant contributions of therapies incorporating virtual reality as a rehabilitation method. Here, we considered the game environment, gamification features, and the generation of emotions.

**Immersion in therapies:** This corresponds to the level of incorporation of the patient in the therapeutic process, managing to adopt therapy as a game and not as a conventional medical process.

**Influence on quality of life:** This involves determining the sources, focusing on improving the living conditions related to muscular-type dystrophies because many of these injuries make it difficult to perform daily tasks in patients who suffer from them, reducing autonomy. The focus of this study is primarily on children and adolescents.

## 3. Results

The results describe a safe, positive, and feasible experience, although obtained primarily after the application of a single session. ADAS software identified 5298 events in total; meanwhile, each student reviewed 4 h and 21 min of footage, obtaining 2033 events. The events were processed based on their time spans, resulting in 11,852 matches. All patients reported little to no discomfort using the headsets during therapy. On average, it took 166 ± 80 s to successfully find a position where both headsets did not cause pain, the user was able to see the screen correctly, and at least four of the primary sensors were green.

The following heat maps allowed us to identify which classes had the most amount of values and predictions while also finding possible confusions between those. The possible values are from 1 to 5 because we used the Manikin’s five regions.

Overall efficiency without VR was 28% and with VR 44%. Eighty-seven percent of the patients reported not being motivated to attend therapy, which also resulted in missing many appointments. One-hundred percent mentioned that their primary objective in attending rehabilitation was to improve their quality of life, but the biggest challenge was sustaining the physical effort and discomfort. Furthermore, using a VR headset works as a distraction from the medical environment, but also encourages one to keep exploring.

## 4. Discussion

As shown in Table 2, there was a clear difference in how much time it took this setup between males and females, and the main reason is the length and amount of hair. If the subject has a ponytail, hair wraps, curly hair, or something similar instead of a straight haircut, it is much more challenging to move the hair around to get the sensors to touch the skin directly. Some studies have even suggested braiding techniques and new types of electrodes to address this [41]. This situation also causes issues during the session because some hairs can crawl under some sensors and cause interruptions. Another detail to consider is that the natural movement from RAGT results in displacing the EEG headset from the original position and rotation, but having the VR device can help maintain those parameters. During the study, parents and patients mentioned that the lack of motivation resulted in attending late and shortening the length of the rehabilitation, causing them to add more sessions to their recovery plan. Patients in the first rehabilitation sessions had more difficulties performing effective work, but had motivation; those in their last sessions had experience making an effective effort, but lacked incentive. Furthermore, small children have very diverse head shapes, so finding the proper position and angle for every headset becomes troublesome. We found that in 75% of the patients that had a sensor lacking proper pressure on the skin for an adequate connection of the EEG, placing the head strap of the virtual reality headset on top of that sensor would add enough pressure. Based on the weighted average of precision, recall, and F1-score from Table 3, Table 4 and Table 5, it is evident that dominance is the best-recognized state, followed by pleasure and arousal. Interestingly, the lowest average was 89%, making it sufficiently reliable. The heat map from Figure 4 shows that for pleasure, there was an 11.7% erroneous classification of the label “unsatisfied”, while “pleased” was 18.75%. Furthermore, it showed that the current experience of RAGT is mainly unsatisfying. Figure 5 displays a similar scenario for arousal with 10.15% of wrongful classification for “dull”, but a more alarming 21.56% for “wide-awake” with incorrect predictions placed on three other regions. On the contrary, the best scenario is in Figure 6 for dominance, where only one label, “powerlessness”, was almost exclusively used with an error of 10.28%. The most common labels were “unsatisfied”, “wide-awake”, and “powerlessness”. This matches the sentiment expressed by the therapists, patients, and their families; it seems that for them, the most critical challenge is to change the lack of satisfaction and dominance. Further research is required to identify which other challenges could be more critical overall.

In previous studies, the Emotiv Insight has been suggested to be used more for daily use rather than research [33], but considering these results, this headset could also be used for research and medical purposes when working with regions instead of high-precision data and if the weight, price, and consistency over long periods are necessary factors. If that is not the case, we suggest using a more robust headset such as the Emotiv Epoc+ or others considered medical-grade [9]. During the study, we found that some of the interpreted values, such as meditation, had a shorter range and variation for the patients with more neurological affectations. Furthermore, these subjects were non-verbal, and it was harder to read their affective state by looking at their facial and body expressions. Roshdy et al. considered it possible to use an EEG to identify affective states in people with neurological disorders [42]. Our findings corroborate that study, that it is possible, but future research projects should consider that those patients will require more calibration since they cannot reach some regions of the SAM.

Furthermore, introducing virtual reality showed an increase of 16% in the efficiency of the session, and this was accomplished with a static scenario. Roosta et al. found that subjects who have elements of a game suited to their motivation type can have a significant difference in motivation and engagement when compared to those who had randomly assigned elements [43]. Therefore, we predict that using a dynamic feedback-based scenario can provide better efficiency than a static scenario. Voigt et al. successfully manipulate PAD parameters by first inducing high arousal and low pleasure using a horror game and then introducing a nature visualization to attain low arousal and high pleasure [27]. Among the limitations we found was the number of participants in the study. Usually, a CRIT center has around eight patients daily that require gait therapy. Some of them avoid muscle atrophy of the lower limbs that they cannot control, and others can have extreme reactions to contact or visual stimuli, reducing the number of eligible users.

### Future Work

The current study shows great promise even with a limited number of patients. The next step is to develop a dynamic feedback-based scenario that uses engagement, boredom, frustration, and meditation to modify the parameters of a VR scenario and compare the results to a static system and one without virtual reality. Furthermore, we should validate the result consistency following the rehabilitation across multiple sessions and discard the usual excitement that users can feel when trying something new.

## 5. Conclusions

This study proposed a method for increasing the efficiency of RAGT sessions for patients with neurological disorders by combining affective data from an EEG, a VR headset for displaying an immersive interaction, and a feedback system. The method combines the use of a Lokomat as the RAG, Emotiv Insight as the EEG, and an Oculus Go for VR. Our preliminary study showed that it is possible to use an EEG headset on patients from 5 to 13 years of age, identify the affective state of the patient with a weighted average precision of 97.5%, recall of 87.9%, and F1-score of 92.3% in general, and correctly placing the EEG and VR headsets while the subject is using a Lokomat. It is essential to consider any obstructions to place the sensors of the EEG device, hair being one of the most complex and time-consuming to overcome. Using a virtual reality device can boost efficiency by 16% more, and the results show great promise. This method allows feedback to the therapist in real-time, even if the patient is non-verbal and has a limited amount of facial and body expressions. Hence, the future work is to develop a dynamic feedback-based system that provides consistency in enhancing efficiency across multiple sessions.

## Figures and Tables

**Figure 1 ijerph-19-09523-f001:**
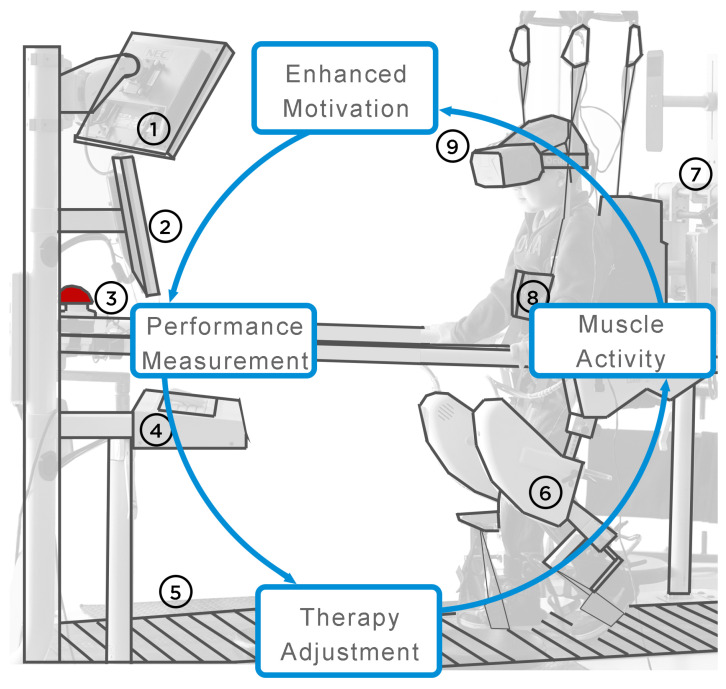
Elements of the system: (1) feedback for the user; (2) feedback for the therapist; (3) emergency stop button; (4) keyboard for controlling Lokomat; (5) treadmill; (6) robotic legs; (7) computer that controls the Lokomat; (8) vest with harness for supporting weight; (9) VR and EEG headsets.

**Figure 2 ijerph-19-09523-f002:**
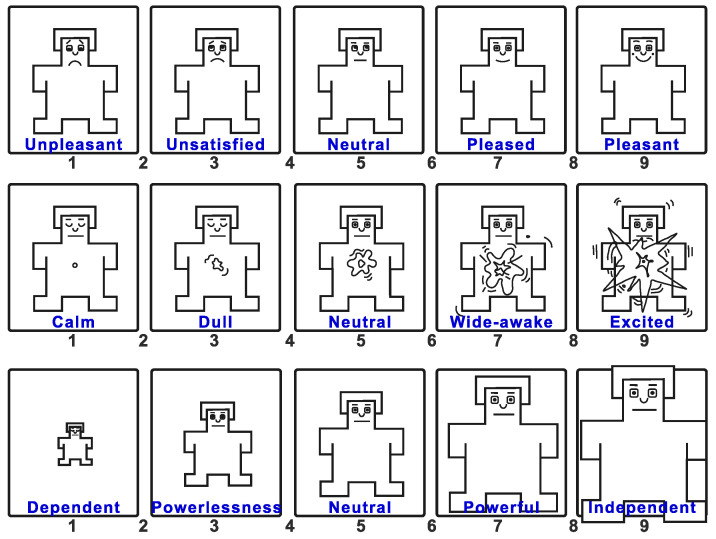
Image presented to reviewers to determine PAD based on observation.

**Figure 3 ijerph-19-09523-f003:**
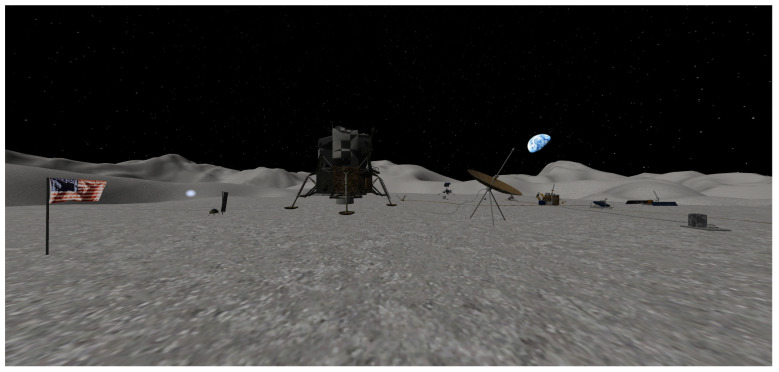
Screenshot of the environment from the VR camera.

**Figure 4 ijerph-19-09523-f004:**
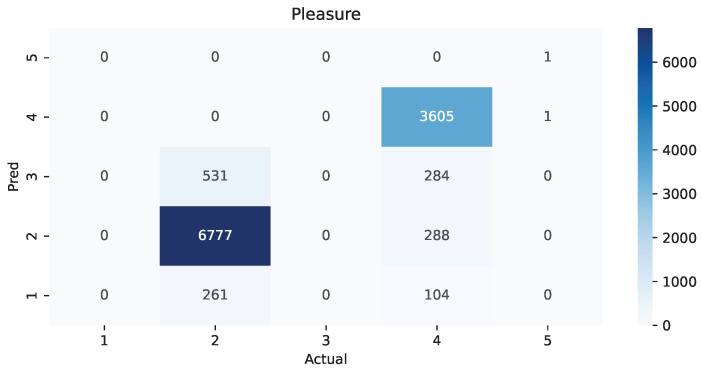
Heat map of the value pleasure comparing actual versus predictions.

**Figure 5 ijerph-19-09523-f005:**
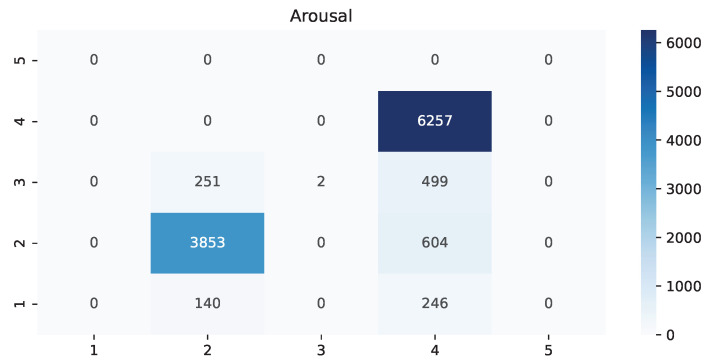
Heat map of the value arousal comparing actual versus predictions.

**Figure 6 ijerph-19-09523-f006:**
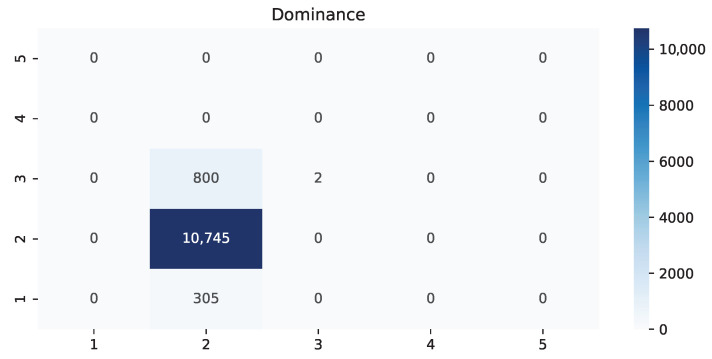
Heat map of the value dominance comparing actual versus predictions.

**Table 1 ijerph-19-09523-t001:** Distribution of videos between students to avoid bias.

Reviewer	Video 1	Video 2	Video 3	Video 4	Video 5	Video 6	Video 7	Video 8
1	AGS1	AGS2	AGS3	AGS4	GDL1	GDL2	GDL3	GDL4
2	AGS2	AGS3	AGS4	GDL1	GDL2	GDL3	GDL4	AGS1
3	AGS3	AGS4	GDL1	GDL2	GDL3	GDL4	AGS1	AGS2
4	AGS4	GDL1	GDL2	GDL3	GDL4	AGS1	AGS2	AGS3
5	GDL1	GDL2	GDL3	GDL4	AGS1	AGS2	AGS3	AGS4
6	GDL2	GDL3	GDL4	AGS1	AGS2	AGS3	AGS4	GDL1
7	GDL3	GDL4	AGS1	AGS2	AGS3	AGS4	GDL1	GDL2
8	GDL4	AGS1	AGS2	AGS3	AGS4	GDL1	GDL2	GDL3

**Table 2 ijerph-19-09523-t002:** Time statistics in seconds of the 3 main activities in each session.

Patient	Sex	Headsets Placing	Session	Unmounting	Total
1	Male	88 s	2075 s	150 s	2313 s
2	Female	188 s	2696 s	77 s	2961 s
3	Male	111 s	1640 s	91 s	1842 s
4	Male	129 s	1996 s	100 s	2225 s
5	Male	75 s	1716 s	118 s	1909 s
6	Female	233 s	1513 s	74 s	1820 s
7	Female	194 s	1655 s	86 s	1935 s
8	Female	310 s	2378 s	153 s	2841 s
Average		166 s	1959 s	106 s	2231 s
Std Dev		80 s	412 s	31 s	451 s
Median		159 s	1856 s	96 s	2080 s

**Table 3 ijerph-19-09523-t003:** Pleasure’s confusion matrix report.

Pleasure				
Region	Precision	Recall	F1-Score	Support
1	0.000000	0.000000	0.000000	0.000000
2	0.959236	0.895363	0.926199	7569.000000
3	0.000000	0.000000	0.000000	0.000000
4	0.999723	0.842093	0.914163	4281.000000
5	1.000000	0.500000	0.666667	2.000000
accuracy	0.876055	0.876055	0.876055	0.876055
macro avg	0.591792	0.447491	0.501406	11,852.000000
weighted avg	0.973867	0.876055	0.921808	11,852.000000

**Table 4 ijerph-19-09523-t004:** Arousal’s confusion matrix report.

Arousal				
Region	Precision	Recall	F1-Score	Support
1	0.000000	0.000000	0.000000	0.000000
2	0.864483	0.907870	0.885645	4244.000000
3	0.002660	1.000000	0.005305	2.000000
4	1.000000	0.822640	0.902691	7606.000000
5	0.000000	0.000000	0.000000	0.000000
accuracy	0.853189	0.853189	0.853189	0.853189
macro avg	0.466786	0.682627	0.448410	11,852.000000
weighted avg	0.951305	0.853189	0.896436	11,852.000000

**Table 5 ijerph-19-09523-t005:** Dominance’s confusion matrix report.

Dominance				
Region	Precision	Recall	F1-Score	Support
1	0.000000	0.000000	0.000000	0.000000
2	1.000000	0.906751	0.951095	11,850.000000
3	0.002494	1.000000	0.004975	2.000000
4	0.000000	0.000000	0.000000	0.000000
5	0.000000	0.000000	0.000000	0.000000
accuracy	0.906767	0.906767	0.906767	0.906767
macro avg	0.334165	0.635584	0.318690	11,852.000000
weighted avg	0.999832	0.906767	0.950936	11,852.000000

## Data Availability

The data used and/or analyzed during the current study are available from the corresponding author upon request.

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
