# Peer review of "Affective States and Virtual Reality to Improve Gait Rehabilitation: A Preliminary Study"

_ijerph, 2022, doi:10.3390/ijerph19159523_

Round 1

Reviewer 1 Report

This study offers an interesting approach to gait rehabilitation involving affective states and VR. As a preliminary study, this paper contains a wealth of useful information and is thought-provoking for the future. However, the reviewer found the problems listed below. I believe the paper would be even better if these could be improved.

 (1) The description in the "2.4 Data Analysis"

It seems that some of the resulting information is mixed in with the method section. Are the numbers in the first three lines of "2.4 Data Analysis" not results? If so, please move this description to the Results section and explain the analysis procedure here.

 (2) Configuration of "Fig. 4-6 Heat Map"

Heat maps may be useful in listing predictive and actual relationships. This time it's explained that there was a missing area and the class could have been non-used or unexpected, but it's confusing. Especially in Figure 4, the lack of 3 in the Actual column can confuse the reader. Since there was no value, it doesn't matter if it is "0", so why not put a temporary value and arrange it in 5x5?

In addition, regarding the heat map, I felt it would be better to align the directions of the Actual axis and the prediction axes. Since PAD has an intensity, this is not a normal x-y graph, but how about a representation in which the intensity of both increases as one moves away from the origin (lower left corner of the map)? By changing the figure in combination with the above 5x5 array, the relationship between the two factors will be easier to understand and the comparison between the figures will be easier.

Author Response

We would like to thank the reviewers and editors for their detailed analysis of the manuscript. Please see the attachment.

Reviewer 2 Report

I am reviewing the article “Affective States and Virtual Reality to Improve Gait Rehabilitation: A Preliminary Study”. The manuscript under consideration is an interesting article on an important topic. However, there are a few major concerns.

1. How did the authors determine the sample appropriate size? Why not power calculations performed a priori? Please provide all parameters for the sample size calculation in the Methods.

2. Basic information such as height, weight and BMI of participants is missing. Please add them.

3. Please add detailed reasons why students ages 19-22 participated.

4. Please explain more details about the Self-assessment Manikin.

5.The authors state that the patients were not motivated, but did they assess their self-esteem and quality of life beforehand?

6. "We found it somewhat effective to position the Virtual Reality headset so that the straps would help some of the sensors to make contact with the skin." This consideration is vaguely worded and it is not clear to what extent it is effective.

7. "This corroborates the sentiment expressed by the therapists, patients, and their families; the most critical challenge found in this study is to change the lack of satisfaction and dominance." I believe this consideration can be strengthened by combining it with an appropriate psychological evaluation. This being said, please soften the wording a bit more.

8."This was accomplished with a static scenario; therefore, we predict that using a dynamic feedback-based scenario will provide even better efficiency." The rationale for this consideration should be explained further.

9.  Since there are many abstract considerations, I would like to see more evidence-based considerations.

Author Response

(The authors gave the same response as above.)

Round 2

Reviewer 2 Report

The authors have carefully made the changes as requested. The manuscript is ok for acceptance.